# Machine Learning Quantification of Intraepithelial Tumor-Infiltrating Lymphocytes as a Significant Prognostic Factor in High-Grade Serous Ovarian Carcinomas

**DOI:** 10.3390/ijms242216060

**Published:** 2023-11-07

**Authors:** Jesús Machuca-Aguado, Antonio Félix Conde-Martín, Alejandro Alvarez-Muñoz, Enrique Rodríguez-Zarco, Alfredo Polo-Velasco, Antonio Rueda-Ramos, Rosa Rendón-García, Juan José Ríos-Martin, Miguel A. Idoate

**Affiliations:** 1Department of Pathology, Virgen Macarena University Hospital & School of Medicine, University of Seville, 41009 Seville, Spain; jmachuca94@hotmail.com (J.M.-A.); felixconde@telefonica.net (A.F.C.-M.); alucmu@gmal.com (A.A.-M.); enriquerodriguezzarco@gmail.com (E.R.-Z.); rosarendon93@gmail.com (R.R.-G.); jjrios@us.es (J.J.R.-M.); 2Gynecology Department, Virgen Macarena University Hospital & School of Medicine, University of Seville, 41009 Seville, Spain; apolov@sego.es; 3Oncology Department, Virgen Macarena University Hospital & School of Medicine, University of Seville, 41009 Seville, Spain; antonioruedaramos@gmail.com

**Keywords:** ovarian cancer, tumor-infiltrating lymphocytes, digital quantification, algorithms, machine learning

## Abstract

The prognostic and predictive role of tumor-infiltrating lymphocytes (TILs) has been demonstrated in various neoplasms. The few publications that have addressed this topic in high-grade serous ovarian carcinoma (HGSOC) have approached TIL quantification from a semiquantitative standpoint. Clinical correlation studies, therefore, need to be conducted based on more accurate TIL quantification. We created a machine learning system based on H&E-stained sections using 76 molecularly and clinically well-characterized advanced HGSOC. This system enabled immune cell classification. These immune parameters were subsequently correlated with overall survival (OS) and progression-free survival (PFI). An intense colonization of the tumor cords by TILs was associated with a better prognosis. Moreover, the multivariate analysis showed that the intraephitelial (ie) TILs concentration was an independent and favorable prognostic factor both for OS (*p* = 0.02) and PFI (*p* = 0.001). A synergistic effect between complete surgical cytoreduction and high levels of ieTILs was evidenced, both in terms of OS (*p* = 0.0005) and PFI (*p* = 0.0008). We consider that digital analysis with machine learning provided a more accurate TIL quantification in HGSOC. It has been demonstrated that ieTILs quantification in H&E-stained slides is an independent prognostic parameter. It is possible that intraepithelial TIL quantification could help identify candidate patients for immunotherapy.

## 1. Introduction

Ovarian cancer is the fifth leading cause of cancer-related death in women. Currently, the standard treatment for high-grade serous ovarian carcinoma (HGSOC) is cytoreductive surgery followed by adjuvant chemotherapy. Although 65% of patients present a good response to chemotherapy, approximately 80% of the tumors recur within the first 10 years [1,2]. An increased antioxidant activity in tumor cells is implicated in tumor resistance to platinum-derived drugs [3]. The recent introduction of new treatments such as bevacizumab and inhibitors of the poly-ADP-ribose polymerase (iPARP) enzyme have improved disease-free survival and overall survival [1,4,5,6].

Despite the advances, the 5-year survival of patients with HGSOC is 44%, impelling the search for new treatments. Immunotherapy trials on ovarian carcinoma have yielded contradictory results, although a significant benefit has been observed in certain patient groups [2,7,8]. There is, therefore, a need to identify new response prediction biomarkers that allow for the proper selection of patients [3,8,9]. Studies have described the important role played by the tumor microenvironment in the response to immunotherapy, focusing mainly on the study of tumor-infiltrating lymphocytes (TILs) and various markers such as programmed death ligand 1 (PD-L1) [9,10,11].

The prognostic impact of TILs has been shown in ovarian carcinoma, with a direct correlation between the number of TILs and patient prognosis [12,13]. Only three studies have examined the predictive/prognostic role of TILs in ovarian carcinoma using hematoxylin-eosin (H&E)-stained slides, with differing results [14,15,16]. The main problem is that TIL quantification was performed using the semiquantitative scoring method recommended by the International TILs Working Group, which suffers from subjectivity and a lack of reproducibility.

In recent years, the boom in digital analysis systems based on artificial intelligence and/or machine learning has led to the development of new TIL counting methods based on a quantitative method that is fast and highly reproducible [11,17].

Due to the lack of TIL quantification studies using digital analysis in H&E-stained slides in HGSOC, we speculated that it might be possible to build an analysis algorithm that can determine its predictive and/or prognostic importance.

## 2. Results

### 2.1. The ieTILs Create an Interesting Phenomenon of Intracordal Tumor Colonization, Which Is Associated with the Neoadjuvant Therapy

Morphologically, we observed an intense infiltrate of TILs in close proximity to the tumor cells, such that the TILs were inserted into the epithelial cords, forming dense inflammatory aggregates that ate away and dissected these tumor cords (Appendix A). We have named this phenomenon “tumor cord colonization” by TILs, a histopathological finding observable in 26.3% (20/76) of the cases. The patient group with tumors colonized by immune cells presented an OS of 66 months and a PFI of 23 months, which was slightly longer than that of the other patients.

The phenomenon of tumor cord colonization by ieTILs was more frequent in the tumors that had been treated with neoadjuvant therapy, which represents 37.5% (12/32) in these tumors versus 18% (8/44) in the rest of the tumors.

### 2.2. There Is a Significant Association between High ieTILs and sTILs Values with OS and PFI

The question we subsequently asked ourselves is whether the TILs could delay the onset of tumor recurrence and, consequently, whether this effect could increase survival (Table 1 and Appendix A).

The patient group with ieTILs-rich tumors (>0.2) presented a longer PFI than the patient group with low-ieTILs tumors (median 29 vs. 12 months; *p* = 0.0004), which implies a recurrence risk 2.5-fold higher for the second group (*p* = 0.0007). Similarly, the patients with high ieTILs presented longer overall survival than the low-ieTILs group (51.1% vs. 12.3%, *p* = 0.01), with the second group having a 2.4-fold higher mortality risk (*p* = 0.01) (Figure 1A,B).

Similarly, the stratification by sTILs led to the creation of two risk groups. The high-sTILs (>1100) patient group presented a longer PFI (median of 27 months) than the low-sTILs group (median of 14 months) (*p* = 0.04), such that the second group had a 1.73-fold greater tendency to recurrence (*p* = 0.04). Similar differences were found in terms of OS, such that the low-sTILs group presented a median OS of 46 months, with a survival rate of 14.2% compared with 45.3% for the high-sTILs group (*p* = 0.038), which resulted in a 2-fold greater mortality risk (Figure 1C,D).

### 2.3. The Role of TILs as Prognostic Biomarkers Is Especially Relevant for Patients Who Have Not Undergone Neoadjuvant Therapy

Given that neoadjuvant therapy is administered to patients in whom the disease has greater dissemination and thus greater difficulty in achieving complete surgical resection, the next objective was to clarify whether there was an additive antitumor effect from the neoadjuvant therapy and immune response.

When grouping the patients using both parameters (neoadjuvant therapy and ieTIL concentration), significant differences were observed in the OS (*p* = 0.01) (Figure 2A) and PFI curves (*p* = 0.01) (Figure 2B). The patient group with high-ieTILs who did not undergo neoadjuvant therapy showed longer survival and a lower tendency of tumor recurrence (Table 1). Similar results were obtained for the sTILs; the high-sTIL group who had not undergone neoadjuvant therapy showed longer survival (*p* = 0.02) (Appendix A) and a longer interval without tumor recurrence, although the association was not statistically significant (*p* = 0.053) (Appendix A). When stratifying (according to ieTILs and sTILs) the patients who underwent neoadjuvant therapy (*n* = 32), no significant differences were observed in either the PFI or OS (Appendix A).

### 2.4. Although the ieTIL Concentration Was Not Higher in the Tumors Mutated for Gene BRCA, This Immune Component Appeared to Be More Efficient in the Tumors in Which the Mutation Was Present

Another interesting aspect was the relationship between TILs and the oncological parameters studied, depending on the neoplasm’s mutational state. As previously explained, we assumed that the *BRCA* gene mutation is associated with increased tumor neoantigenicity and, consequently, a greater immune response.

Twenty-four of the 76 tumors (31.6%) presented a somatic mutation of gene *BRCA*. There were no significant differences in the PFI and OS when comparing the patients with mutated *BRCA* tumors and those with non-mutated tumors. Based on this gene’s mutational state, there were also no significant differences in the quantity of ieTILs (0.21 vs. 0.19, *p* = 0.6) or sTILs (1334.12 vs. 1056.11, *p* = 0.2).

In the patients with high ieTIL concentrations, there were no significant differences in the clinical behavior according to the tumor’s mutational state. The patient group whose tumors showed a *BRCA* gene mutation and a high ieTIL concentration presented a significantly longer OS and PFI than the patients with mutated tumors and low ieTIL concentrations. The patient group with non-mutated tumors and low ieTIL concentrations showed significant differences only in the PFI (hazard ratio (HR) 2.44, *p* = 0.007) (Figure 2C, Table 1).

These differences in clinical parameters were smaller or were not observed for the patients with non-mutated *BRCA* tumors regardless of the ieTIL concentration (Appendix A, Appendix A). There was also statistical significance for the sTILs exclusively in the patients with mutated *BRCA* tumors for the PFI (Table 1 and Appendix A).

### 2.5. The Patients with Both a Complete Surgical Resection and High Tumor ieTIL Concentration Had an Especially Favorable Prognosis

Achieving a complete resection is one of the fundamental prognostic factors in HGSOC. Therefore, we proposed the possible additive prognostic factor of the immune infiltrate depending on the surgery’s efficacy.

When stratifying the patients according to the degree of surgical resection, the most important finding was that the patient group with complete surgical resection and a high ieTIL concentration had a significantly better prognosis, both in terms of OS (*p* = 0.0005) and PFI (*p* = 0.008) (Figure 3A,B).

### 2.6. The ieTIL Concentration Was an Independent Prognostic Factor Both for OS and PFI in the Multivariate Analysis

In the multivariate Cox regression analysis for OS, both complete surgical resection of the tumor (HR 3.28, *p* = 0.002) and high ieTIL concentration (HR 2.94, *p* = 0.02), were independent favorable prognostic factors. Interestingly, the results were similar for the PFI, but the prognostic impact of the ieTIL concentration exceeded that of complete surgical resection (HR 2.95, *p* = 0.001 vs. HR 1.99, *p* = 0.02). Neoadjuvant therapy was also a prognostic factor for OS (HR 2.95, *p* = 0.004) but not for PFI (HR 1.57, *p* = 0.09). In contrast, the mutational state of *BRCA* and the sTIL values were not correlated with any of the oncological reference parameters (Figure 4).

## 3. Discussion

### 3.1. The Importance of TIL Quantification According to Its Relationship with Tumor Cords

The investigation into TILs in HGSOC initially focused on ieTILs, relegating the role of sTILs to a secondary level, given that the former showed prognostic significance [13,18,19,20,21]. In contrast, however, published studies have indicated that sTILs, which also have prognostic significance in HGSOC [15,16,22], constitute an equally relevant prognostic factor that is easier to assess [21,23,24]. However, although both types of TILs should be evaluated and quantified to have a more complete understanding of the antitumor immune response, ieTILs should have special consideration because they reflect the immune component that establishes close contact with the tumor cells.

Moreover, using the term “intratumoral” to refer to the infiltrate that is inserted in the epithelial cords of the neoplasm creates a problem with term precision [21,24]. The term “intraepithelial” seems more appropriate for this immune component. Also, the concept of intraepithelial (ieTILs) needs to encapsulate all those cells that are inside the tumor cords and those that are present in the epithelial–stromal tumor interface. This statement is based on the demonstrated fact that the effect of tumor lysis of T cells requires direct contact with the tumor cells [14,25].

In our opinion, ieTILs can be differentiated from sTILs in most cases of HGSOC. By proceeding in this manner, we established the prognostic role for ieTILs and sTILs, although the former component plays a more relevant role in the antitumor response. It is also highly important to establish a ratio between the number of TILs and tumor cells. We, therefore, introduced a new method for assessing this phenomenon. The relevance of the antitumor immune response should be evaluated based on the quantity of neoplastic cells that need to be eliminated. Therefore, the ratio of the number of ieTILs to the number of tumor cells for a specific area provides a more accurate idea. There is no doubt that the concept of quantity needs to be accompanied by the concept of ieTIL and sTIL functionality and its relationship with the inhibitory factors secreted by the tumor cells or by the contact with immunosuppressant macrophages [14], among others.

### 3.2. The Importance of ieTILs as an Independent Prognostic Factor

There are few publications that have covered the prognostic role of TILs in HGSOC; only three have quantified TILs in H&E-stained preparations [14,15,16]. Our study’s results are especially relevant because it is the first study to quantify TILs in H&E in HGSOC using digital image analysis with an integrated automatic learning system, which has enabled us to avoid the frequent intraobserver and interobserver variability that occurs in studies employing semiquantitative evaluation, as well as achieving greater accuracy. We can, therefore, conclude that the ieTIL parameter is a relevant independent prognostic factor, unlike sTILs. These results agree with those published by James et al. [14] but not with those of authors who concluded that only sTILs constitute an independent prognostic factor [15,16]. When comparing our results with other published results, we must consider that these studies conducted the TIL evaluation by following a semiquantitative and manual procedure [21], which detracts from objectivity.

### 3.3. Neoadjuvant Therapy and BRCA Gene Mutation Appear to Influence the Prognostic Effect of TILs

We have shown a significant association between TILs and PFI in patients who did not undergo neoadjuvant therapy but not in patients who underwent this therapy. The published conclusions are conflicting on this point [4,16,26,27,28,29]. To understand this lack of correlation, at least two aspects need to be considered: (1) the starting condition of patients who have undergone neoadjuvant therapy is that of a greater degree of tumor dissemination, which could be associated with an immune exhaustion condition and tumor overgrowth; and (2) that the post-therapy ieTIL assessment needs to take into account the possible effect of interference of the therapy on the quantity and/or function of immune cells and, ultimately, on their antitumor efficacy [22]. These two effects could explain why the neoadjuvant therapy behaved as an adverse prognostic factor in our results. To resolve these hypotheses, we would need immune cell functionality studies and to precisely understand other interesting aspects, such as the ratio of the immune effector response/immunosuppressive immune response.

With regard to the *BRCA* gene mutation, it is conceivable that this mutation could a priori exert a certain boosted effect on the immune response in HGSOC [12,30,31,32,33]. In our study, however, there was no increase in immune infiltrate in the tumors with somatic mutation of the *BRCA* gene or prognostic differences between the two groups. However, we observed that the ieTILs have a prognostic role in the tumors with the *BRCA* gene mutation, which does not occur in the tumors without the mutation. Therefore, this beneficial effect of the ieTILs could be explained not by the quantity of infiltrate but rather by the efficiency of the immune response, such that this was greater in the context of the mutation. To make conclusions about this point, we would need to characterize the tumor microenvironment more precisely.

### 3.4. The ieTILs as Prognostic Modifier in Tumor Resection Groups

Complete tumor cytoreduction is one of the most relevant independent prognostic factors in HGSOC [1,15], as confirmed in our study, which concluded that complete surgical resection is associated with a longer increased PFI and increased OS. We have shown that the prognosis is clearly superior for the patients in whom there is complete cytoreduction and high ieTIL concentrations, which supports the finding that ieTIL concentration is a factor of special prognostic significance. The prognostic impact that complete cytoreduction exerts is similar to that of the ieTILs, a highly interesting finding that represents a new perspective on the importance of the antitumor effect of ieTILs. It can, therefore, be postulated that the assessment of ieTILs could permit a relatively rapid classification of patients with HGSOC into different prognostic groups. This important prognostic impact of ieTILs was not observed for sTILs, which supports our hypothesis of a lesser role for this tumor component in the immune response.

### 3.5. The Predictive Role of TILs, Especially ieTILs, and the Potential Application as an Immunotherapy Biomarker

To date, studies on the use of immunotherapy in ovarian carcinoma have not achieved the expected results [4,7,8,9]. However, in the American Society of Clinical Oncology (ASCO) Congress of 2023, the results of a multicenter study were presented that observed that patients treated with chemotherapy, bevacizumab, and durvalumab and with maintenance therapy of bevacizumab, durvalumab, and olaparib showed longer disease-free survival than patients with standard treatment, regardless of the mutational state of the *BRCA* gene [34]. These findings open a new therapeutic option for patients with ovarian carcinoma, with the recommended use of biomarkers that help predict which patients will benefit especially from these treatments for a cost-effective use of available resources.

There is a demonstrated relationship between TILs and the response to immunotherapy, such that in most neoplasms the tumors with a greater quantity of TILs have a better response [9,10,11,35]. The low success of immunotherapy in ovarian cancer has made TILs a seldom relevant parameter, although the observance of poorer response to treatment of HGSOC with a relatively low quantity of TILs suggests a certain role for the immune infiltrate in antitumor resistance [36]. To resolve this issue, studies are needed that approach TIL quantification using digital analysis that facilitates the objectivity and reproducibility of the results. Although we have not examined the relationship between TILs and immunotherapy, our results indicate that in the group with HGSOC with high ieTILs concentrations, there is a certain antitumor braking effect, as observed for example in its association with the development of long-term tumor relapses. We can, therefore, postulate a possible synergistic effect of TILs with immunotherapy.

### 3.6. Considerations of the Study’s Methodological Limitations

The relatively low number of patients in our study is one of its main limitations when drawing conclusions. However, our results have shown the usefulness of the digital quantification of TILs in H&E-stained preparations, both intraepithelial and stromal. We are aware that our algorithms need to be validated in larger patient series.

The difficulties inherent in the evaluation and classification process for tumor and immune cells cannot be minimized. These difficulties were encountered especially in the patients with HGSOC who underwent neoadjuvant therapy, due to a greater difficulty in quantifying the immune infiltration of tumor cords, given that the definition of the tumor epithelium–stroma interface is not as clear in this clinical context. This complication is due to the fact that the algorithm employed to differentiate the two components is based on a system of distinguishing pixels. The main trait employed for classification is the color of the pixels that compose the image, such that a considerable concentration of immune cellularity in the tumor epithelium–stroma interface complicates the classification process. Despite these drawbacks, this methodology represents a more accurate and reproducible measurement than semiquantitative manual counting systems. To achieve greater accuracy, we would need methodological optimization and validation of our results with new technologies such as multiplex immunofluorescence, which leads to greater accuracy in quantifying immune cell populations.

We can, therefore, conclude that digital analysis with machine learning provides a more accurate quantification of TILs in HGSOC, classifying them according to their location. Our results allow us to postulate that the ieTIL concentration could constitute an independent prognostic biomarker, both of OS and PFI, in HGSOC. Lastly, the recent developments in immunotherapy as a treatment in HGSOC have opened the door to the search for new predictive biomarkers that allow for patient selection for these treatments. The ieTIL concentration quantified by machine learning in H&E-stained preparations is a promising immune biomarker.

## 4. Material and Methods

This is a retrospective observational study of 76 patients diagnosed with HGSOC between 2013 and 2021 in our hospital. The requirements for inclusion were a diagnosis of stage III or IV HGSOC, treatment with cytoreductive surgery, a follow-up of at least 10 months, and a breast cancer gene (*BRCA*) mutation in the neoplasm. This last requirement was performed using next-generation sequencing using the Illumina platform.

The patients’ mean age was 59.2 years (range 36–83 years). The median platinum-free interval (PFI), defined as the time elapsed between the start of treatment and the diagnosis of the first tumor recurrence, was 19 months (range 0–80 months). The median overall survival (OS) was 61 months (range 10–134 months).

All patients were treated with cytoreductive surgery combined with 6 cycles of adjuvant chemotherapy, undergoing neoadjuvant therapy in certain cases. Depending on whether they underwent neoadjuvant therapy, the patients were divided into two groups. The first group comprised those patients who underwent the therapy (42% (32/76) of the cases) and in whom the surgical specimen was studied after the neoadjuvant therapy. The second group consisted of the patients who did not undergo neoadjuvant therapy, representing the remaining 58% (44/76) and in whom the surgical specimen was studied before starting the therapy. The patients selected for neoadjuvant therapy presented a higher degree of neoplasm dissemination at diagnosis, with 44% of the patients in stage IV, compared with 13.6% for those who did not undergo the therapy, with a median OS of 46 months and a PFI of 13 months. The more relevant clinical-demographic data have been included in Table 2.

After the cytoreductive surgical treatment, complete resection was achieved in 63% of the cases, with a rate of 59.4% for the patients who underwent neoadjuvant therapy versus 65.9% for those who did not undergo the therapy.

The H&E-stained sections of each patient were scanned with the Ventana HT360 scanner (Roche Diagnostics, Tucson, AZ, USA) with a 400× magnification (resolution of 0.25 microns per pixel) and stored as TIF files. After scanning, a pathologist verified the quality of all digital preparations to ensure that all regions were correctly focused.

All digital analyses were performed using QuPath v3.2.0 (https://qupath.github.io/ (accessed on 15 October 2021)), an open-source software platform for digital image analysis, with image analysis algorithms using integrated automatic learning (Appendix A). First, two independent pathologists selected two representative areas of 2 mm^2^ from each of the preparations. Preprocessing was then applied to define the color channels, with the aim of homogenizing the staining of the various preparations to facilitate the subsequent analysis. Representative training areas were then selected to segment the tumor epithelium (red) and stroma (green). The pixel training algorithm was applied to produce the tissue classification, with additional training regions to improve the accuracy of the classification in the epithelium and stroma. The pixel classifier or pixel classification algorithm was then applied to all representative areas of 2 mm^2^ in each digital slide, which were then reviewed by a pathologist. For the proper application of the pixel classification algorithm, we had to overcome various problems with incorrect classification mainly due to the enormous tumor heterogeneity characteristic of high-grade serous carcinoma. This heterogeneity is manifested in the form of various morphological patterns of presentation (papillary, micropapillary, solid, transitional, etc.), variability in tumor morphology (clear cell, dark cells, fusiform, rounded, etc.), and certain subtle changes in the tumor secondary to neoadjuvant therapy, such as a certain stromal fibrosis and changes in the morphology of fibroblasts or tumor cells, imperceptible to the human eye but identifiable with imaging software (QuPath v0.3.2). We, therefore, had to initially develop (always following the same methodology) two-pixel classifiers, one for patients who had undergone neoadjuvant therapy and another for those who had not. After applying these two differentiated algorithms, a highly accurate classification was achieved in approximately 70% of cases. After this, we successively developed (following the same methodology) new pixel classifiers until a correct classification was achieved in all cases. After this, two separated regions were created for “stroma” and “epithelium”.

We then applied the cell detection to each region with the following characteristics, thereby starting the creation of the cell classification algorithm or cell classifier detection image, haematoxylin OD; requested pixel size, 0.5 μm; bottom radius, 8 μm; mean radius of the filter, 0 μm; sigma, 1.5 μm; minimum cellular area, 10 μm^2^; maximum cellular area, 400 μm^2^; threshold, 0.1; maximum intensity of bottom, 2. A pathologist performed the quality control of the cellular detection. To classify the cells detected in the tumor cells, immune cells (TILs), and stromal cells, among others (false detections, bottom), we used neural networks with eight hidden layers (maximum iterations, 100). To assist the algorithm in performing an accurate classification, we also added the characteristics of softened objects with radii of 25 μm and 50 μm to supplement the existing measures of individual cells. We used a set of 785 annotated images to train the system, observing, although to a lesser extent, cell classification errors due to the reasons listed above. The sequential training of the cell classifier algorithm was, therefore, repeated, which, after several rounds, allowed us to correctly classify the cells as “tumor”, “stromal”, or “immune” in the 76 cases (Figure 5). A pathologist performed the quality control of the algorithm for classifying the detected cells.

After quality control, we exported the data, which included for each case two annotations, each of which was divided into “epithelium” and “stroma”. From each of them, we obtained a series of parameters that included the total surface area of the stroma and tumor epithelium and the quantification of the total number of cells detected, which included the total number of tumor epithelial cells, stromal cells, and immune cells.

We obtained a mean of the results of the two annotations for each case. These raw data were analyzed to obtain the various parameters that enabled us to assess the quantification of the immune infiltrate:-Intraepithelial TILs (ieTILs) = ratio of the total number of immune cells in close contact with the tumor epithelium/total number of tumor cells.-Stromal TILs (sTILs) = total number of immune cells in the stromal compartment/surface area of the tumor stroma (mm^2^).

After obtaining these results, we performed a statistical study to correlate these parameters with others such as age, achievement of complete resection, platinum-free interval, overall survival, application or not of neoadjuvant therapy, and mutational state of gene *BRCA*.

The analyses were carried out using R and SPSS version 29 software. Descriptive statistics were employed, presenting absolute and relative values of qualitative variables using tables. The measures of central tendency, position, and variability were applied to quantitative variables. The normality assumptions of the quantitative variables were assessed using the Shapiro–Wilk test.

In inferential statistics, bivariate analyses were performed to examine clinical characteristics in relation to mortality and recurrence. For normally distributed quantitative variables, the independent samples *t*-test was employed. In cases of non-normality, the non-parametric Mann–Whitney test was utilized. For qualitative variables, the Chi-square test or Fisher’s exact test (for expected frequencies < 5) was applied.

Survival analysis was conducted, comparing Kaplan–Meier curves for mortality and relapse events based on ieTILs and sTILs values. This was accomplished using the Log Rank test (Mantel-Cox) or the Generalized Wilcoxon test. Cox regression with non-parametric adjustment for proportional hazards was also employed. Key R libraries used for analysis included library(survival), library(ggsurvfit), library(gtsummary), and library(ggplot2). Statistical significance was established at a *p*-value < 0.05.

## Figures and Tables

**Figure 1 ijms-24-16060-f001:**
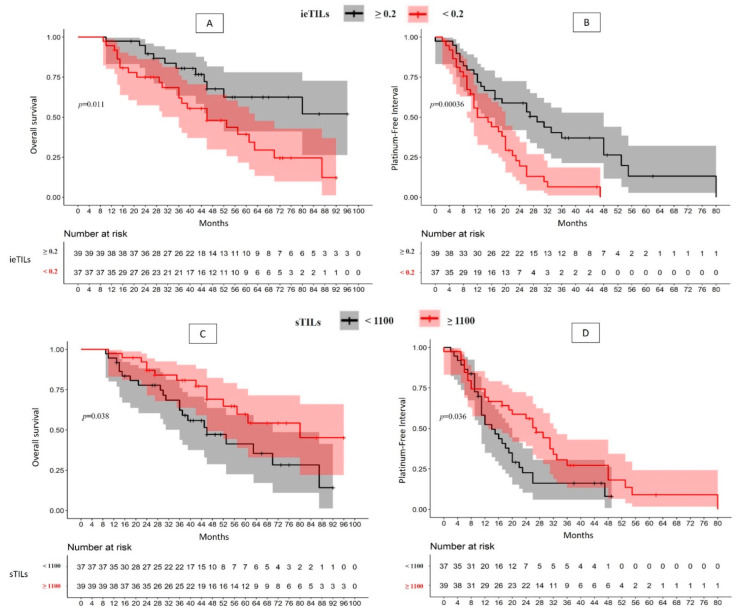
(**A**–**D**) Photocomposition. Overall survival and platinum-free interval of the patients with high-grade serous ovarian carcinoma stratified by the quantity of ieTILs ((**A**,**B**), respectively) and by the quantity of sTILs ((**C**,**D**), respectively).

**Figure 2 ijms-24-16060-f002:**
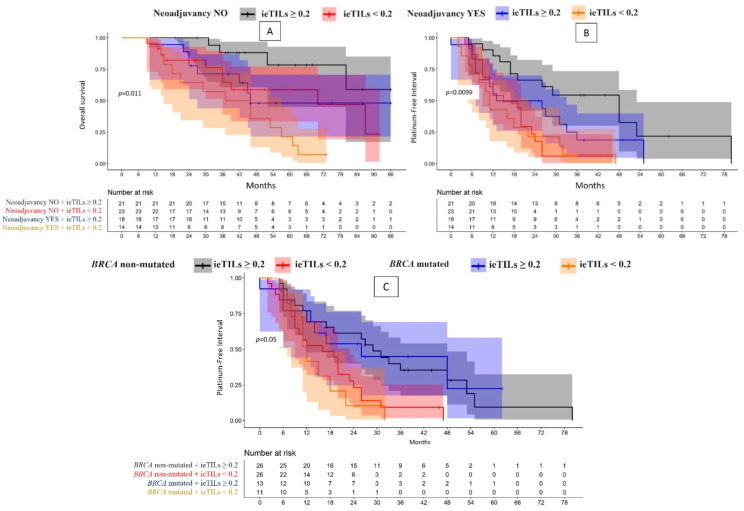
(**A**–**C**) Comparative study of overall survival and platinum-free interval in patients with high-grade serous ovarian carcinoma based on whether they received neoadjuvant treatment and the number of ieTILs ((**A**,**B**), respectively), and of the platinum-free interval-based on *BRCA* mutational status and ieTIL concentration (**C**).

**Figure 3 ijms-24-16060-f003:**
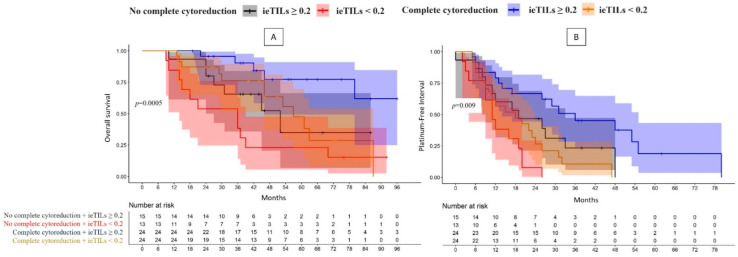
(**A**,**B**) Complete resection. Comparative curves of the overall survival and platinum-free interval of the patients with high-grade serous carcinoma according to whether complete resection was achieved and the ieTIL concentration ((**A**,**B**), respectively).

**Figure 4 ijms-24-16060-f004:**
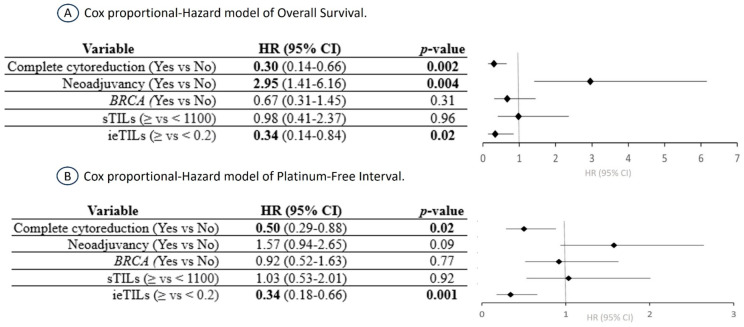
Cox proportional-Hazard model of overall survival and platinum-free interval in high-grade serous ovarian carcinoma.

**Figure 5 ijms-24-16060-f005:**
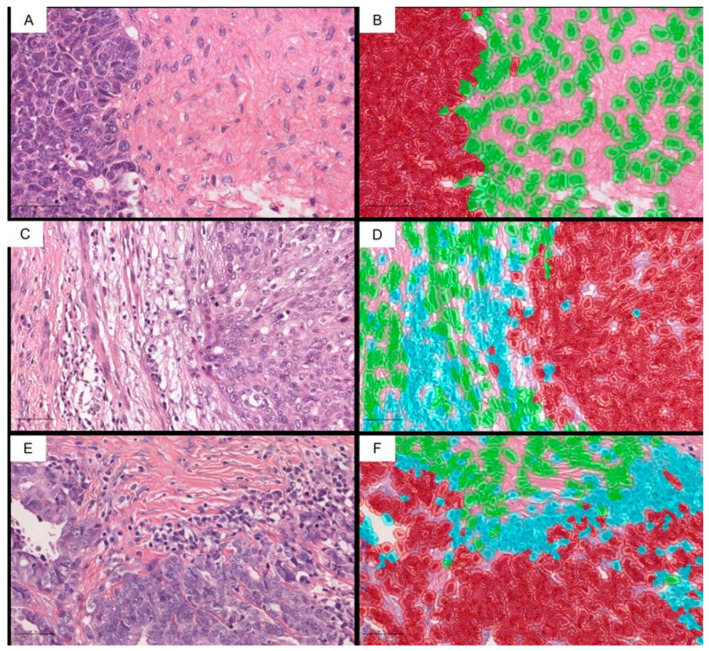
Examples of applying the cell classification algorithm, with the tumor cells in red, the immune cells in blue, and the stromal cells in green. (**A**,**B**) Immune desert neoplasia, where accompanying immune cellularity has not been identified. (**C**,**D**) Carcinoma in which there is an accompanying immune infiltrate of stromal predominance (sTILs), although there are some lymphocytes touching the tumor epithelium. (**E**,**F**) There is an immune infiltrate in close contact with the tumor epithelium in which there are areas of lymphocytes intermixing with the tumor epithelium (intracordal). (HE; 200×).

**Table 1 ijms-24-16060-t001:** Comparative analysis of overall survival and platinum-free interval stratified by intraepithelial TILs and stromal TILs in different patient subgroups.

				Overall Survival			Platinum-Free Interval		
				% survivors (median)	*p*	HR (95% CI)	Median	*p*	HR (95% CI)
Overall (76)		ieTILs	High	51.1%	***p =* 0.01**	**2.40** (1.19–4.84); *p =* 0.01	29 m	***p =* 0.0004**	**2.59** (1.50–4.50); *p =* 0.0007
			Low	12.3% (46 m)			12 m		
		sTILs	High	45.3% (80 m)	***p =* 0.04**	**2.02** (1.02–3.99); *p =* 0.04	27 m	***p =* 0.04**	**1.73** (1.02–2.93); *p =* 0.04
			Low	14.2% (46 m)			14 m		
Neoadjuvant therapy	YES (32)	ieTILs	High	48.1% (46 m)	***p =* 0.01**	2.88 (0.86–9.58); *p =* 0.1	13 m	***p =* 0.01**	2.09 (0.97–4.48); *p =* 0.06
			Low	7.14% (37 m)		**6.67** (2.15–20.76); *p =* 0.001	11 m		**4.09** (1.83–9.16); *p =* 0.0006
	NO (44)		High	58.8%		1	48 m		1
			Low	23.5% (70 m)		2.93 (0.92–9.37); *p =* 0.07	15 m		**3.45** (1.60–7.45); *p =* 0.002
	YES (32)	sTILs	High	27.4% (46 m)	***p =* 0.02**	**4.99** (1.39–17.91); *p =* 0.01	13 m	*p =* 0.053	1.78 (0.91–5.03) *p =* 0.09
			Low	12.8% (45 m)		**7.53** (2.05–27.70); *p =* 0.002	11 m		**2.89** (1.78–9.83); *p =* 0.01
	NO (44)		High	63%		1	29 m		1
			Low	21.2% (70 m)		**4.38** (1.22–15.75); *p =* 0.02	15 m		**2.68** (1.22–5.91); *p =* 0.01
Somatic *BRCA*	Mutated (24)	ieTILs	High	68.8%	*p =* 0.1	1	26 m	***p =* 0.05**	1
			Low	0% (53 m)		**4.16** (1.09–15.88); *p =* 0.04	12 m		**3.08** (1.37–6.92); *p =* 0.006
	Non-mutated (52)		High	45.2% (80 m)		1..09 (0.33–3.05); *p =* 0.8	29 m		1.01 (0.46–2.32); *p =* 0.99
			Low	17.7% (46 m)		1.93 (0.84–4.42); *p =* 0.1	16 m		**2.44** (1.28–4.63); *p =* 0.007
	Mutated (24)	sTILs	High	63%	*p =* 0.2	1.40 (0.47–4.23); *p =* 0.6	26 m	*p =* 0.22	0.83 (0.38–1.81); *p =* 0.6
			Low	21.2% (70 m)		2.75 (0.96–7.84); *p =* 0.6	12 m		**2.69** (1.17–6.20); *p =* 0.02
	Non-mutated (52)		High	55.9%		1	27 m		1
			Low	22% (46 m)		2.10 (0.92–4.82); *p =* 0.08	17 m		1.42 (0.76–2.64); *p =* 0.3
Complete cytorreduction	YES (48)	ieTILs	High	61.8%	***p =* 0.0005**	1	36 m	***p =* 0.009**	1
			Low	0% (57 m)		**3.05** (1.07–8.71); *p =* 0.04	18 m		**2.99** (1.44–6.21); *p =* 0.003
	NO (28)		High	34.9% (52 m)		**3.27** (1.03–10.42); *p =* 0.04	19 m		2.10 (0.95–4.65); *p =* 0.07
			Low	15.4% (36 m)		**6.27** (2.17–18.09); *p =* 0.0007	11 m		**5.29** (2.31–12.11); *p =* 0.00008
	YES (48)	sTILS	High	47.5% (80 m)	***p =* 0.009**	1	29 m	*p =* 0.1	1
			Low	0% (63 m)		1.78 (0.68–4.63); *p =* 0.2	15 m		1.41 (0.70–2.86); *p =* 0.3
	NO (28)		High	38.9% (52 m)		2.35 (0.78–7.07); *p =* 0.1	19 m		1.35 (0.59–3.06); *p =* 0.5
			Low	20.7% (36 m)		**3.48** (1.48–7.07); *p =* 0.004	12 m		**2.55** (1.30–4.99); *p =* 0.006

HR: Hazard Ratio; ieTILs: intraepithelial tumor-infiltrating lymphocytes; sTILs: stromal tumor-infiltrating lymphocytes; m: months; CI: confidence interval.

**Table 2 ijms-24-16060-t002:** Relevant clinical–demographic characteristics of the patients included in the study.

Age (Years)	36–83 (Median 59.2)
FIGO stage	IIIA: 7; IIIB: 14; IIIC: 35 IVA: 8; IVB: 12
Neoadyuvancy treatment	32/76 (42%)
*BRCA* mutation	24/76 (31.6%)
Overall survival (months)	10–134 (median 61)
Platinum-free interval (months)	0–80 (median 19)

## Data Availability

The datasets used and/or analysed during the current study are available from the corresponding author upon reasonable request.

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
