# Peer review of "Machine Learning Quantification of Intraepithelial Tumor-Infiltrating Lymphocytes as a Significant Prognostic Factor in High-Grade Serous Ovarian Carcinomas"

_ijms, 2023, doi:10.3390/ijms242216060_

Round 1
Reviewer 1 Report
Comments and Suggestions for Authors
The present Article, ijms-2685419, entitled: (MACHINE LEARNING QUANTIFICATION OF INTRAEPI-THELIAL TUMOR-INFILTRATING LYMPHOCYTES AS A SIGNIFICANT PROGNOSTIC FACTOR IN HIGH-GRADE SEROUS OVARIAN CARCINOMAS)
The Research article is very informative and written in a very concise way. However, I have the following minor comments:
- - p value need to be italic along the whole manuscript.
- -Numbers below figures (2-4) cannot be seen.
- -Authors need to follow journal format to quite references inside text and in the journal list.
- - Is it possible to state the precision % of the applied method???
Comments on the Quality of English LanguageMinor editing of English language required
Reviewer 2 Report
Comments and Suggestions for Authors
The manuscript is very interesting but presents some points that must be improved. Figures must be improved while tables are clear. My comments are listed below:
Introduction: It deserves to be pointed out that ovarian cancer recurrence is mainly due to the occurrence of chemotherapy resistance. In fact, cancer cells show an increased antioxidant responce (see PMID: 37175546 ) allowing a protection from platinum-derived drugs.
Figure 2 and 3: Images quality is very low and must be improved
Figures: remove the figure number (e.g. "Figure 1" from the image since it is already written in the legend
A table with the clinico-demographic characteristics of patients must be added
An accurate revision of typing errors is recommended
Round 2
Reviewer 2 Report
Comments and Suggestions for Authors
the manuscript has been significantly improved and can be accepted in the present form